# Analysis of Primary Liquid Chromatography Mass Spectrometry Data by Neural Networks for Plant Samples Classification

**DOI:** 10.3390/metabo12100993

**Published:** 2022-10-19

**Authors:** Polina Turova, Andrey Stavrianidi, Viktor Svekolkin, Dmitry Lyskov, Ilya Podolskiy, Igor Rodin, Oleg Shpigun, Aleksey Buryak

**Affiliations:** 1Faculty of Chemistry, M.V. Lomonosov Moscow State University, 1-3 Leninskie Gory, Moscow 119991, Russia; 2A.N. Frumkin Institute of Physical Chemistry and Electrochemistry, Russian Academy of Sciences, 31-4 Leninsky Prospect, Moscow 119071, Russia; 3BostonGene Corporation, University Office Park III, 95 Sawyer Road, Waltham, MA 02453, USA; 4Faculty of Biology, M.V. Lomonosov Moscow State University, 1-12 Leninskie Gory, Moscow 119234, Russia; 5Bruker Ltd., Pyatnitskaya 50/2 Build. 1, Moscow 119017, Russia

**Keywords:** Apiaceae, raw LC-MS data, neural networks, support vector machine, augmentation

## Abstract

Plant samples are potential sources of physiologically active secondary metabolites and their classification is an extremely important task in traditional medicine and other fields of research. In the production of herbal drugs, different plant parts of the same or related species can serve as adulterants for primary plant material. The use of highly informative and relatively easily accessible tools, such as liquid chromatography and low-resolution mass spectrometry, helps to solve these tasks by means of fingerprint analysis. In this study, to reveal specific plant part features for 20 species from one family (Apiaceae), and to preserve the maximum information content, two approaches are suggested. In both cases, minimal raw data pretreatment, including rescaling of time and *m*/*z* axes and cutting off some uninformative regions, was applied. For the support vector machine (SVM) method, tensor unfolding was required, while neural networks (NNs) were able to work directly with squared heatmaps as input data. Moreover, five data augmentation variants are proposed, to overcome the typical problem of a lack of data. As a result, a comparable F1-score close to 0.75 was achieved by SVM and two employed NN architectures. Eight marker compounds belonging to chlorophylls, lipids, and coumarin apio-glucosides were tentatively identified as characteristic of their corresponding sample groups: roots, stems, leaves, and fruits. The proposed approaches are simple, information-saving and can be applied to a broad type of tasks in metabolomics.

## 1. Introduction

Herbal extracts contain dozens and even hundreds of endogenous secondary plant metabolites [1]. The strategy that is commonly applied to control the authenticity and quality of such products includes HPLC-UV quantification of the main constituents (quality markers). However, due to the complexity of their chemical composition, a more comprehensive approach that uses “chromatographic fingerprints” has also been employed [2]. Due to the extent of the variability among the same type of plant material, the fingerprints should be informative enough to allow distinguishing samples from different groups [1]. Such fingerprint chromatograms are accepted by several regulatory authorities, including the WHO and FDA [3,4]. The performance of a chromatographic fingerprint depends on the separation degree and concentration distribution of the chemical components; therefore, its information content is limited. Thus, two-dimensional (2D) fingerprinting has been introduced to better address the chemical complexity of the herbal preparations [5,6]. Moreover, hyphenated chromatographic and spectrometric approaches, such as HPLC-DAD [7,8,9,10,11,12,13,14], GC-MS [15,16], and HPLC-MS [17,18], can greatly enhance the potential of chromatographic fingerprinting. However, in this case, one has to operate with raw analysis data in 3D space.

The first obvious way to deal with such complex data is to reduce it to a simple chromatographic fingerprint. This can be done by two approaches. First, by selecting target peaks at several wavelengths and removing the baseline differences, thus obtaining a combined chromatogram as a result [7,8,9]. The second option is to select several most-inhabited wavelength signals and concatenate them into one fingerprint vector with rejection of the initial retention time (RT) axis [10]. A more sophisticated way is to perform fusion profiling and extract as many relevant peaks as possible [11]. As an alternative, several chromatographic fingerprints at different wavelengths can be evaluated separately [12], but this is impractical in view of further statistical data analysis.

The second way is to work with a whole three-dimensional HPLC fingerprint spectrum represented as an image [13,14]. In this case, unconventional for analytical chemistry, image-processing techniques can be introduced to conduct qualitative or quantitative analysis. For example, different image moments, including Chebyshev moments, were employed to assess ginsenoside contents in *P. notoginseng* [13] and to classify Pudilan Xiaoyan tablet samples from several pharmaceutical companies [14].

Similar to DAD, a mass spectrometric detector provides an additional dimension for HPLC fingerprints and also provides an opportunity to identify the compounds with the most characteristic peaks. In case of MS detection, total ion current (TIC) chromatograms are often employed to construct simple 1D fingerprints [15,17]. This, however, does not lead to a prominent increase in informativity. Alternatively, a number of chromatographic peaks with different RT and base *m*/*z* signals can be extracted as a new feature space, which has now also become the most widespread approach in plant metabolomics studies [18]. Such peak tables can even be fused with similarly treated HPLC-DAD data to construct combined fingerprints with increased information content [16].

Building up an *m*/*z*—RT peak table for a whole dataset is not a trivial task and requires sophisticated mathematical algorithms to be employed to deal with the retention time shift, noise signals, and artifacts [19]. Therefore, the aforementioned image approach can be applied to extract the unique features directly from raw LC-MS or GC-MS data, bypassing this highly complex preprocessing step. In this case, supervised machine learning methods can be applied to provide a basis for sample discrimination. To our best knowledge, there has been no attempt regarding direct pattern recognition for plant material samples; however, this strategy has been employed to distinguish different types of cancer [20,21].

Convolutional neural networks (CNNs), shallow neural networks (NNs), and support vector machines (SVMs) have been employed to learn patterns directly from the GC-MS abundance matrix [20]. The authors of this study applied an original approach, dividing the initial data into small segments that are sufficiently large to include a volatile chemical’s entire peak. Each segment included abundances for all 411 *m*/*z* values and was represented as a grayscale or RGB image for one-channel and three-channel input, respectively. To increase the robustness of the training, data augmentation was applied by shifting along the RT dimension and randomly increasing abundancies in the 10% range. CNN demonstrated higher performance with respect to NN and SVM, which detected a high number of false positives. SVM can perform better if more preprocessing is applied to the data [16]. A 3D CNN was successfully applied in the analysis of raw (“unassigned”) HPLC-MS data for cancer phenotype classification [21]. Each file was represented as 98 images with dimensions of 512 × 512 pixels with color-coded intensities, thereby achieving a classification accuracy for several types of cancer of slightly above 90%.

There are plenty of examples where tissue-specific secondary metabolites have been found for one plant by analyzing untargeted MS data [22,23,24]; however, the number of studies where such differences are found for plant parts from different species is limited. NMR-based metabolomic analysis was applied to distinguish the roots/rhizomes and aerial parts of three Actaea species [24]. The application of such fingerprint methods is important since in quality control of botanical products, different plant parts can be considered as adulterants of the main material [24,25].

In the previous study [26], we compared the application of several unsupervised methods, including parallel factor analysis (PARAFAC), principal component analysis (PCA), independent component analysis (ICA), non-negative matrix-factorization (NMF), and unsupervised feature selection (UFS), for treatment of HPLC-MS data with minimal preprocessing. In the extracts from leaves of 19 plants belonging to the Apiaceae family, the characteristic coumarin markers and other secondary metabolites, which may serve as potential quality markers, were selected. In this work, we studied the possibilities of using convolutional NN with cross entropy as a loss function, Siamese NN (SNN) with triplet loss function, and SVM in the processing of HPLC-MS fingerprints. The developed approach was applied to distinguish different plant parts from the Apiaceae family. Residual convolutional NN was chosen as the basic architecture for applied NN as they had already been used for mass-spectrometry data [21,27].

## 2. Experimental

### 2.1. Materials and Reagents

HPLC-grade acetonitrile from Panreac (Barcelona, Spain) and MeOH > 99.9% purity from Burdick & Jackson (Muskegon, Ml, USA) were used. Formic acid > 99.9% purity was purchased from Acros (Geel, Belgium) and ammonium formate ≥ 99.0% purity was obtained from Sigma-Aldrich (Steinheim, Germany). Deionized water was prepared with a Milli-Q system from MilliporeSigma (Burlington, MA, USA).

### 2.2. Sample Preparation

Plant material was collected in Iran, Portugal, Kyrgyzstan, and Uzbekistan in 2013–2019 and housed in Moscow University Herbarium (MW) and partially in the private collection of Dmitry Lyskov. Apart from additional parts of 19 plants studied earlier [26], it included fruits and roots of *Prangos chelantofolia*, two new plant samples of *Bilacunaria microcapra*, and one additional plant sample of *Ferulago phialocarpa*. Each specimen was separated into its available parts (roots, leaves, stems, and fruits) according to their physiological shape. Thus, the final set of samples contained 11 roots, 15 stems, 20 leaves, and 16 fruits and flowers (Table 1). All samples were dried and ground to a fine powder for analysis. The fine powder was weighed (10 mg), suspended in 1 mL of 75% (*v*/*v*) methanol, and ultrasonically extracted for 30 min at 50 °C. All extracts were prepared in triplicate. The obtained extracts were centrifuged, and then 10-times diluted with 10% aqueous acetonitrile prior to HPLC-MS analysis.

### 2.3. Instrumentation

The samples were analyzed by using a Thermo Scientific Dionex Ultimate 3000 (Waltham, MA, USA) system with an AB Sciex Qtrap 3200 (Concord, ON, Canada) mass spectrometer. A C18 column (Acclaim RSLC 2.1 × 150 mm, 2.2 μm) was used to perform the chromatographic separation of 5 μL of each sample injected into a gradient system at a flow rate of 350 μL/min. The oven temperature was 35 °C The mobile phase consisted of 0.5% formic acid in water (A) and acetonitrile (B). The starting eluent was 10% B. Its proportion was held constant for 3 min, increased linearly to 95% from 3 to 20 min, held constant at 95% until 22 min, returned to the initial composition (10% B) at 22.2 min, and then held constant for 4.8 min to re-equilibrate the column.

The mass spectrometer was operated in positive ion mode and set to the total ion chromatogram (TIC) mode (100–1200 *m*/*z*). The optimized MS conditions were as follows: capillary voltage of 5500 V, declustering potential of 40 V, entrance potential of 10 V, source temperature of 350 °C, nebulizing gas pressure of 30 psi, drying ion source gas pressure of 40 psi, and curtain gas pressure of 15 psi.

HPLC-HRMS analysis was performed on a Bruker Elute LC system coupled on-line with a Bruker Impact II high-resolution Quadrupole Time-of-Flight Instrument. HPLC separation was conducted on a C18 column (Intensity Solo 2.1 × 100 mm, 1.8 µm) at a gradient flow rate (from 0.200 to 0.480 mL/min). Two solvents were used: (A) 5 mM ammonium formate and 0.01 % FA in a MeOH:H_2_O 1:99 mixture; and (B) 5 mM ammonium formate and 0.01% FA in MeOH. The gradient was as follows: 0–0.1 min 4% B; 0.1–1 min linear gradient from 4 to 18.3%; 1–2.5 min linear gradient from 18.3 to 50% B; 2.5–14 min linear gradient from 50 to 99.9% B; 14–16 min 99.9% B; 16–16.1 min linear gradient from 99.9 to 4% B; 16.1–20 min 4% B.

### 2.4. Software and Packages

Raw LC-MS files from the Sciex instrument were converted into mzXML format using MSConvert from ProteoWizard Tools. Data analysis was performed in Python 3 using the following modules: pymzML for mzML data files parsing [28]; pytorch for convolutional neural networks architecture creation and pytorch-metric-learning for SNN implementation [29]; scikit-learn for SVM and performance metrics [30]; matplotlib [31] and seaborn [32] for data visualization; OpenCV [33] for data augmentation; and optuna [34] for the Bayesian optimization algorithm.

### 2.5. LC-LRMS Data Treatment

The model dataset consisted of 186 samples; that is, three replicates each of the 62 plant samples, representing 7 genera from the Apiaceae family. Apart from the additional parts of 19 plants studied earlier [26], it also included fruits and roots of *Prangos chelantofolia*, two new plant samples of *Bilacunaria microcapra*, and one additional plant sample of *Ferulago phialocarpa*. All samples were analyzed in the same chromatographic conditions by HPLC-LRMS. The composition of the mobile phase varied over a wide range (10–95% of acetonitrile) during the gradient program in order to elute both polar and less-polar compounds.

For time axis unification, linear interpolation [35] was used, and the time step size was chosen as 0.03 min. The final time range was from 1.5 to 24 min. For the mass axis unification, the intensities for signals with residual masses in the range from −0.35 to +0.65 were summed and assigned to a cell with the corresponding integer *m*/*z* value. Data from all samples were combined into one tensor with dimensions of 186 × 750 × 1200, which corresponds to the number of samples, the number of points on the retention time scale, and the number of *m*/*z* values, respectively. All implemented algorithms are available at the GitHub repository (https://github.com/turovapolina/CNN-LC-MS, accessed on 19 October 2022).

## 3. Results and Discussion

### 3.1. LC-MS Data Pretreatment and Augmentation

In previous work [26], we explored several possible data preparation methods, including tensor unfolding and tensor decomposition, in combination with unsupervised machine learning methods for clustering plant samples from the Apiaceae family. Application of unsupervised methods allows to identify the most significant characteristic signals corresponding to 23 quality markers or potential chemotaxonomic markers. If the purpose of a study is to identify the unique characteristic markers for different groups of samples, or to classify unknown samples into predetermined groups (by origin, by quality, by age, or by part of the plant), then supervised machine learning methods should be employed. Direct sample classification using methods such as SVM also involves several stages of data preparation. The data tensor should be unfolded to be analyzed by this approach. The unfolding procedure takes a tensor of dimensions I × J × K and rearranges it in such a way that the number of samples, I, remains unchanged, and two other dimensions (*m*/*z* and RT) are combined into a single new dimension, with size J × K. Data vectorization includes the unification of the time and mass axes for all samples, which in the case of low-resolution HPLC-MS data involves rounding raw masses to integer values (see Section 2.5). Thus, a resulting tensor with dimensions of 186 × 750 × 1200 (samples: RT: *m*/*z*) was unfolded into a 186 × 900,000 data matrix prior to SVM analysis.

Raw LC-MS data of each sample can be presented by a heatmap, which could be considered as a medical image (Figure 1). This leads to the idea of different neural networks (NNs) employment for sample classification, as it was done for medical images [27,36]. After the time and mass axes were rescaled, heatmaps with new dimensions and a unified grid were reconstructed for all samples. These heatmaps were cropped to a squared form (750 × 750) via rejection of the less-important signals in low (100–200 *m*/*z*) and high (950–1300 *m*/*z*) *m*/*z* regions. Compared to the approaches described in the literature [20,21], where a series of images was generated for each sample, representation of LC-MS data in the form of a single image seems promising because it saves spatial information and preserves the correlation between the adjoined signals in the sample. The squared images are ideal as input data for a convolutional NN with different losses and architectures.

Augmentation is a standard and effective technique for solving problems such as a lack of data in deep learning. The most common augmentation methods for 2D data samples are stretching, squeezing, rotating, denoising, and cropping [36]. For the HPLC-MS data, augmentation should be used with caution because the axes have different physical meanings, and, for instance, rotation operations cannot be applied. In our work, we suggested several augmentation techniques: stretching and shrinkage along the retention time axis, mass shift, and intensity alteration. A description of the proposed methods of augmentation and the main parameters are presented in Table 2.

All five augmentation procedures, if applied simultaneously to the initial samples, could increase the number of samples in the dataset by 15 times. Separately, each procedure simulates a process that could actually occur during a slight change in the conditions of the HPLC-MS experiment or during experiment transfer to another instrument. For example, relatively small mass shifts in the raw data (Figure 2) could affect the signal intensities that are retrieved after the mass gridding procedure (summing the peaks in the spectrum to integer values).

Thus, the final dataset contained 186 initial images and 2604 samples generated via augmentation, which were further randomly divided into training (70%) and testing (30%) fractions (Figure 1), providing that all three replicates of each plant material extract would be either in the test set or in the training set. The random split procedure was repeated three times (cross-validation) and each time the model was trained and tested on a new random dataset.

### 3.2. Data Analysis Using the SVM

SVM was chosen as a base method for sample classification [16,20]. This method is suitable for the analysis of data with a large number of features due to its simplicity and tendency not to overfit. In addition, it is possible to extract the feature importance per class from the SVM model by using a linear kernel function.

Due to the large variation in chemical composition among the plant samples belonging to the same genus, and thus the strong class imbalance, it was impossible to classify samples in accordance with their taxonomy; indeed, the aim of the study was to test the ability of the developed approach to be employed for plant part differentiation. There were four classes in the dataset: roots, stems, leaves, and fruits/flowers. The F1-score, accuracy, and precision were used for evaluation of the algorithms (Table 3).

The highest accuracy (0.77) was achieved by SVM with full augmentation of the initial dataset, which demonstrates the benefits of using the proposed augmentation procedures for LC-MS data. Moreover, each augmentation procedure applied have not distorted the consistency of the data and, therefore, have not affected the classification accuracy.

### 3.3. Class-Important Features from the SVM Model Results

After classification using the SVM method, an algorithm for extracting the important features per class was proposed. The features’ weight matrix for each class was extracted, features were ranked by weight value, and the first 2000 with the maximum weight were selected. Further, each selected feature from each class was tested for the appearance of the corresponding signal on the chromatograms of samples that do not belong to the considered class. The intensity of the signal at the point with the corresponding coordinates (RT—*m*/*z*) was compared to the average value of the intensities in the mass spectrum at this time point, which approximately corresponded to 7–10 in terms of the S/N ratio. If the signal intensity did not exceed five times the corresponding average intensity value, then it was considered that the peak is absent in the sample. In the final list of unique significant features, isotopic signals with the same retention times as well as signals with the same *m*/*z* at neighboring time points were consolidated.

Via application of the proposed algorithm, eight characteristic signals of the four sample groups were selected. It should be noted that some non-unique class features that could be found at trace levels (S/N above 3 and below 10) in several samples from other groups were chosen. However, the selectivity of the suggested algorithm could be illustrated by the overlaid extracted ion chromatograms (Appendix A). To preliminary annotate the corresponding compounds, HPLC-HRMS and MS/MS experiments (Appendix A) were performed for the selected signals or related precursor ions (see Section 2.3).

Compound **1** possesses a molecular weight of 598 deduced from the protonated molecule ([M + H]^+^) peak at *m*/*z* 599.1967 (C_27_H_35_O_15_, eluted at 9.0 min), which undergoes a successive neutral loss of apiofuranose and glucopyranose, which yields two fragment ion signals at *m*/*z* 467 and 305. The fragment ion at *m*/*z* 203 corresponds to the side chain (C_5_H_10_O_2_) cleavage. The remaining fragmentation pattern (Table 4) was typical for a monosubstituted furanocoumarin [37,38]. The exact position of the sugar chain could not be assigned, and this compound was tentatively identified as Api(f)-Glc-heraclenol or its isomer [39]. Compound **2** has a molecular weight 456 determined by the protonated molecule ([M + H]^+^) signal at *m*/*z* 457.1338 (C_20_H_25_O_12_, eluted at 6.4 min). Similar to compound **1**, the observed fragment ion peaks at *m*/*z* 467 and 305 indicated apiofuranose and glucopyranose cleavage. The fragmentation pattern was in accordance with the one described for apiosylskimmin [40]. The occurrence of such apioglycosides in roots is well-known [39,41].

Compounds **3** and **4** possess molecular weights of 606 and 602, deduced from the protonated molecules ([Mg-M]^+^) at *m*/*z* 607.2184 (C_36_H_31_N_4_O_4_Mg, eluted at 10.4 min) and *m*/*z* 603.2078 (C_33_H_31_N_4_O_6_Mg, eluted at 8.9 min), respectively, which was confirmed by the clear absence of sodium and potassium adducts in this case, and also by the presence of the corresponding clusters [2M + NH_4_]^+^, [2M + Na]^+^, and [2M + K]^+^ [42]. These elemental compositions obviously point to chlorophyll-related species. Further structural elucidation was not possible due to the low abundances of the fragment ions in their ESI/MS^2^ spectra. It should be noted, however, that the relatively high DBE values (i.e., 20–23) are close to those of protochlorophyllide and its analogues found in steams [43]. Moreover, many tissue-specific protochlorophyllides can be found in stems, shoots, and algae [44,45]. Therefore, these two peaks can be considered as potentially novel compounds.

Compounds **5** and **6** possess molecular weights of 624 and 620, deduced from the protonated molecule ([M + H]^+^) ion peaks at *m*/*z* 625.2662 (C_35_H_37_N_4_O_7_, eluted at 21.8 min) and *m*/*z* 621.2738 (C_36_H_37_N_4_O_6_, eluted at 22.3 min), respectively. Although the fragmentation data is limited because of the low intensity of these two peaks, it could be suggested that they are also related to the chlorophyll cycle. By comparison with the MS/MS data from [46], Compound **5** was tentatively identified as 15^1^-hydroxy-lactone-pheophorbide a, and Compound **6** was tentatively identified as methylpheophorbide b.

Compounds **7** and **8** showed protonated molecules that differ by 2 Da and both contain a phosphatidylcholine head group, which was deduced from the predominant fragment ion *m*/*z* 184.0741 (C_5_H_15_NO_4_P^+^). The single fatty acid side chain was observed as a fragment ion at *m*/*z* 339 (and *m*/*z* 337 correspondingly for Compound **8**). Moreover, the ion peak corresponding to the loss of H_2_O was also observed in the MS/MS spectra (Table 4). Thus, Compounds **7** and **8** were tentatively assigned as oleoyl lysolecithin and linoleoyl lysolecithin, respectively. Apparently, these wide-spread lipids cannot serve as unique markers; however, their content in fruit extracts was significantly higher, and their peaks were correctly identified as characteristic variables.

**Table 4 metabolites-12-00993-t004:** Characteristic *m*/*z* features extracted by SVM and the preliminary identified compounds.

N°	Retention Time (*m*/*z*)	[M + H]^+^, *m*/*z* (Δ, ppm) *	Adduct Ions, *m*/*z*	Main MS/MS Fragments, *m*/*z*	Annotation	Reference
1	9.0 (305)	599.1967 (C_27_H_35_O_15_, 0.6)	621.1785 [M + Na]^+^ 637.1546 [M + K]^+^	599−467 = Api(f) 467−305 = Glc 305−269 = 2H_2_O 305−203 = C_5_H_10_O_2_ 203−175 = CO 203−159 = CO_2_ 203−147 = 2CO 159−131 = CO	Coumarin-related signal: Api(f)-Glc-heraclenol (or its isomer)	[39]
2	6.4 (495)	457.1338 (C_20_H_25_O_12_, −1.7)	479.1125 [M + Na]^+^ 495.0900 [M + K]^+^	457−325 = Api(f) 325−163 = Glc 163−119 = CO_2_ 163−107 = 2CO	Coumarin-related signal: Api(f)-Glc-hydroxycoumarin (Apiosylskimmin)	[40]
3	10.4 (607)	607.2184 C_36_H_31_N_4_O_4_Mg (−2.0)	1186.5182 [2M + NH_4_]^+^ 1191.4738 [2M + Na]^+^ 1207.4478 [2M + K]^+^	410.1335 (C_21_H_20_N_3_O_6_, 2.8)	Chlorophyll-related signal: Tissue-specific protochlorophyllide analog	[42]
4	8.9 (603)	603.2078 (C_33_H_31_N_4_O_6_Mg, −2.6)	1178.4983 [2M + NH_4_]^+^ 1183.4536 [2M + Na]^+^ 1199.4277 [2M + K]^+^	440.1416 (C_29_H_18_N_3_O_2_, −0.8)	Chlorophyll-related signal: Tissue-specific protochlorophyllide analog (e.g., Mg-oxo-purpurin-18)	[42,45]
5	21.8 (625, 627)	625.2662 (C_35_H_37_N_4_O_7_, −0.8)	647.2492 [M + Na]^+^ 663.2235 [M + K]^+^	625−607 = H_2_O 621−565 = C_2_H_4_O_2_ 565−537 = CO	Chlorophyll-related signal: (15^1^-hydroxy-lactone-pheophorbide a or its isomer)	[46]
6	22.3 621 (623)	621.2738 (C_36_H_37_N_4_O_6_, 4.8)	643.2533 [M + Na]^+^ 659.2278 [M + K]^+^	621−593 = CO 621−561 = C_2_H_4_O_2_ 561−533 = CO	Chlorophyll-related signal: (Methylpheophorbide b or its isomer)	[46]
7	18.7 (522)	522.3570 (C_26_H_53_NO_7_P, 3.1)	544.3373 [M + Na]^+^	522−504 = H_2_O 522−339 = C_5_H_13_NO_4_P 522−184 = C_21_H_38_O_3_ 184−166 = H_2_O 184−124 = C_3_H_10_N 184−104 = PO_3_ 104−86 = H_2_O	Lipid-related signal: PC (18:1)	[47]
8	17.0 (520, 521)	520.3408 (C_26_H_51_NO_7_P, 1.0)	542.3239 [M + Na]^+^	520−502 = H_2_O 520−337 = C_5_H_13_NO_4_P 520−184 = C_21_H_36_O_3_ 184−166 = H_2_O 184−124 = C_3_H_10_N 184−104 = PO_3_ 104−86 = H_2_O	Lipid-related signal: PC (18:2)	[47]

* [M-H + Mg]+ for Compounds **3** and **4**.

### 3.4. Specificity of Selected Markers

The variability in chemical composition of the investigated plants is very high [26]. Even for different plant parts of these species, the total ion current chromatograms revealed this drastic variation (Appendix A). However, a supervised machine learning approach allowed us to select the relevant signals from the raw data. In Figure 3, the relative abundances for 23 earlier selected characteristic compounds found in the leaves of these plants are compared. Plant part distribution of these compounds shows three main clusters: two of them correspond to the high content of linear monosubstituted furanocoumarins (pabulenol, xanthotoxin, etc.) in leaves and fruits and an increased concentration of angular furanocoumarins, with their pyrano-analogues (cnidiadin, decursin, etc.) in fruits; in turn, the third one has a similar concentration level as the latter coumarin group in roots and stems. In contrast, the distribution of the eight specific markers discovered from the SVM results (see Section 3.3) shows a clear connection with the plant part sample groups (Figure 3B). This demonstrates the ability of the suggested approach to find the minor characteristic features hidden behind the very complex and highly variable composition of plant extracts.

### 3.5. Data Analysis Using the Neural Networks

It is important to note that HPLC-MS data tensor unfolding into a long vector of intensities corresponding to the RT—*m*/*z* pairs, deprives the spatial structure of the data. If all samples in the data are not perfectly aligned, then serious complications will arise during the construction of the model and the quality of the classification may suffer. Convolutional neural networks are able to handle a 2D dataset directly and preserve spatial interactions between adjacent signals by learning the feature representations over small squares of input data. Each compound’s peak is presented by several timepoints and several isotopes, which means that the peak is represented by a small area on the heatmap or by several areas due to unintentional fragmentation or adduct formation happening at the ionization source. Therefore, the original features, which are adjacent on the sample’s heatmap, can potentially belong to one marker compound, and using the convolution procedure, they are combined into higher-order features, which were further used for samples classification. Feature learning occurs throughout the data matrix, allowing objects to be displaced or moved across the matrix and still be detected by the network.

*Classic ResNet architecture with cross entropy as loss function*. The ResNet18 network, which consists of “residual” blocks, was taken as the basic architecture. This architecture was developed in 2015 and has been successfully used for image analysis [36].

The input layer of ResNet18 expect data to have square dimensions, which in our case required cropping of the mass scale from *m*/*z* 200 to *m*/*z* 950 (see Section 2.5). The next step was data normalization. Since the network was pretrained on the ImageNet dataset, LC-MS data were also normalized using the mean and standard deviation of the ImageNet dataset. A weighted sampler was used for the neural network training process and compensated class imbalance; therefore, samples from different classes appeared in a batch at the same frequency.

*SNN with triplet loss function.* In case of the classification problem, when the number of representatives of each class is small or the difference between classes is minimal, the use of SNN is relevant. During training, such a network calculates a similarity function between the objects from the same and different classes. For a model in this task, it was decided to use a triplet loss function. The architecture of this network was also based on the classical ResNet18 architecture, which is a network with 18 convolutional layers. The network was again taken with weights pretrained on the freely available ImageNet dataset. The dimension of the last fully connected layer was changed from 1000 to 32. The margin loss for the SNN was defined using a Bayesian optimization algorithm.

The performances of two NNs’ were compared with the SVM classifier on the fully augmented data. For the controlled conditions, the train and test splits were obtained with fixed random seed. As can be seen from Table 5, both NNs slightly outperformed the SVM method in precision, recall, and F1-score. The best F1-score was achieved for the distinction of leaves and stems samples, most likely because the number of samples with these labels in the training dataset was the biggest. On the contrary, the number of root samples in the training set was relatively small, while samples with the fruits label included fruits, flowers, buds, and other parts of the inflorescence (Table 1). Because of the mentioned reasons, the F1-score for these two classes was lower. Nevertheless, using the developed LC-MS-based approach with application of deep learning tools led to the direct classification of different plant parts from a relatively broad number of studied Apiaceae species.

## 4. Conclusions

A scheme for minimal pretreatment of raw LC-MS data prior to SVM and NN application was suggested. For SVM, a tensor unfolding procedure was applied, while for the NN application, each sample was transformed into a squared image (750 × 750) via rejection of the less-important *m*/*z* regions. This approach was applied in the analysis of different parts of plants in the Apiaceae family. Eight differentiation markers for distinguishing the roots, stems, leaves, and fruits of 20 species in the Apiaceae family were found by SVM analysis of unfolded LC-LRMS data and further tentatively characterized by LC-HRMS. Chlorophyll-related markers were selected for stems and leaves groups, while coumarin apio-glucosides and lipids were chosen for roots and fruits, respectively. Five approaches for LC-MS data augmentation that simulate real experimental deviations, including chromatogram stretching/shrinkage, gradient stretching, mass shifts, and intensity alteration, were suggested, and their application slightly increased the classification accuracy for both SVM and NN. The most accurate results were obtained for stems (0.89) and leaves (0.94) by using SNN with the triplet loss function.

The results obtained herein demonstrate the potential of using direct LC-MS-based classification with application of ML methods for distinguishing plant parts as metabolite sources for further medical use. This approach also can be further extended for classification of any type of object using raw LC-MS data.

## Figures and Tables

**Figure 1 metabolites-12-00993-f001:**
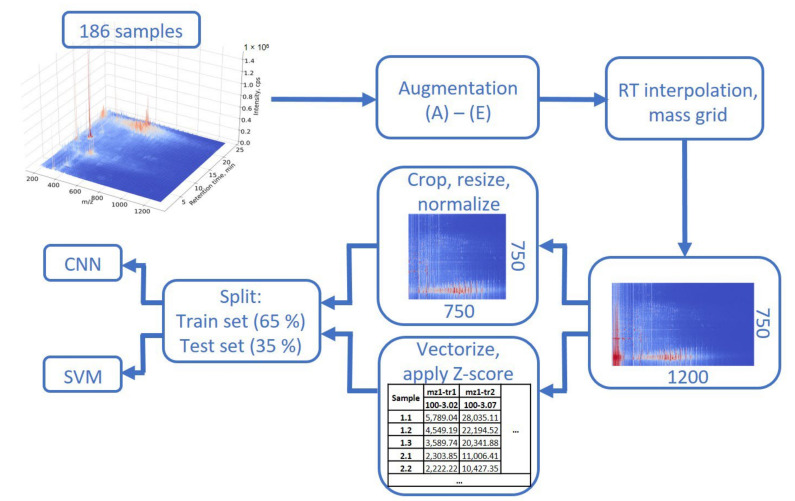
LC-MS data preparation pipeline for CNN and SVM application.

**Figure 2 metabolites-12-00993-f002:**
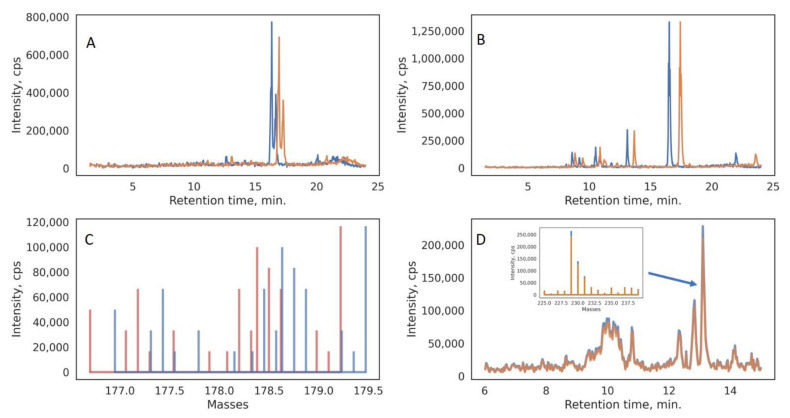
Visualization of the four employed data augmentation techniques: RT shift (**A**); gradient RT shift (**B**); mass shift (**C**); intensity alteration (**D**). Augmented data are shown in orange, whereas original data are shown in blue.

**Figure 3 metabolites-12-00993-f003:**
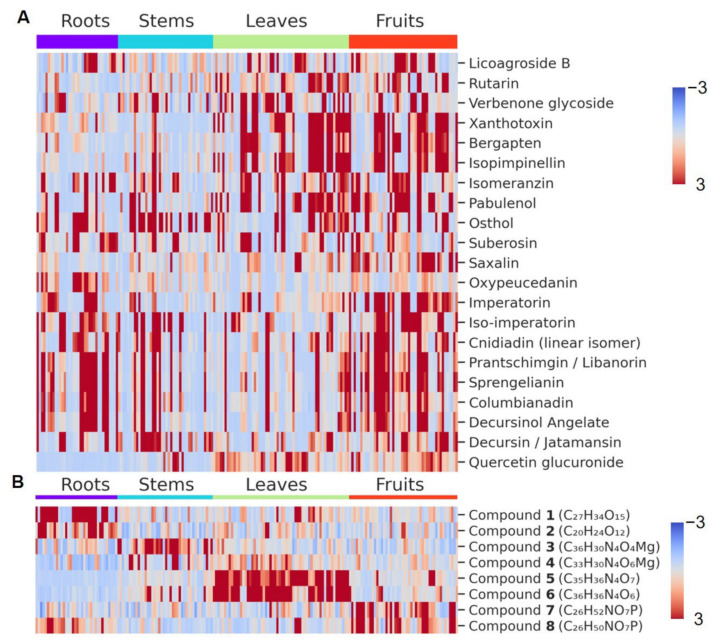
Plant part distribution of coumarins (**A**) and the annotated compounds (**B**).

**Table 1 metabolites-12-00993-t001:** List of investigated specimens: 62 plant parts from 20 different species (Apiaceae).

Species	Plant Parts (#)	Specimen’s Voucher
*Prangos pabularia*	Leaves (1.1), Fruits (1.2), Stems (1.3)	MW0858238
*Ferulago phialocarpa*	Stems (2.1), Leaves (2.2)	031-IR-19
*Cachrys libanotis*	Leaves (3.1), Inflorescence (3.2), Roots (3.3)	MW0798144
*Prangos acaulis*	Leaves (4.1), Roots (4.2), Fruits (4.3)	MW0744005
*Prangos ferulacea*	Stems (5.1), Fruits (5.2)	MW0751912
*Prangos didyma*	Fruits (6.1), Stems (6.2)	MW0857912
*Ferulago subvelutina*	Stems (7.1), Leaves (7.2), Inflorescence (7.3)	098-IR-19
*Prangos ammophila*	Leaves (8.1), Roots (8.2), Inflorescence (8.3)	MW0857867
*Prangos trifida*	Leaves (9.1), Fruits (9.2), Stems (9.3)	MW0798580
*Ferulago angulata*	Stems (10.1), Leaves (10.2), Roots (10.3)	085-IR-19
*Cachrys sicula*	Inflorescence (11.1), Leaves (11.2), Stems (11.3), Roots (11.4)	MW0798143
*Prangos chelantofolia*	Fruits (12.1), Roots (12.2)	MW0753034
*Ferulago contracta*	Leaves (13.1), Stems (13.2)	053-IR-19
*Cachrys pungens*	Fruits (14.1), Leaves (14.2)	MW0784701
*Bilacunaria microcapra*	Roots (15.1), Leaves (15.2), Fruits (15.3), Stems (15.4)	018-IR-19
*Diplotaenia cachrydifolia*	Inflorescence (16.1), Leaves (16.2), Stems (16.3), Roots (16.4)	164-IR-19
*Bilacunaria microcapra*	Leaves (17.1), Roots (17.2), Stems (17.3), Fruits (17.4)	162-IR-19
*Ferulago phialocarpa*	Roots (18.1), Leaves (18.2)	169-IR-19
*Azilia eryngioides*	Roots (19.1), Leaves (19.2), Stems (19.3)	167-IR-19
*Seseli olivieri*	Stems (20.1), Leaves (20.2)	173-IR-19
*Prangos crossoptera*	Fruits (21.1), Leaves (21.2)	MW0753036
*Bilacunaria microcapra*	Leaves (22.1), Inflorescence (22.2)	028-IR-19
*Seseli ghafoorianum*	Stems (23.1), Leaves (23.2)	124-IR-19

**Table 2 metabolites-12-00993-t002:** Description of the procedures employed for data augmentation.

Type of Augmentation	Description	Variation Range (Step Size)	Real Process during Experiment	Augmentation Coefficient
Chromatogram stretching along the entire retention time axis (A)	Each mass-chromatogram was stretched by adding of new time points with intermediate signal intensity values	±30 (10) timepoints *	Wrong pump calibration (incorrect flow rate)	N_samp_ × 6
Gradient chromatogram stretching along the retention time axis (B)	Each mass-chromatogram was split into 15 segments (50 timepoints) and similar stretching procedure was applied to each segment with increasing number of inserted timepoints	From 1 to 7 points for each segment	Wrong gradient program or insufficient flow from organic phase pump	N_samp_
Gradient chromatogram shrinkage along the retention time axis (C)	Each mass-chromatogram was split into 15 segments (50 timepoints) and shrinkage procedure was applied to each segment with increasing number of inserted timepoints	From 1 to 7 points for each segment	Wrong gradient program or insufficient flow from water phase pump	N_samp_
Mass shifts (D)	Each raw *m*/*z* value in each spectrum is shifted to a specific Δ. This Δ is smaller for low masses and bigger for high masses (linear dependence).	(1) From ±0.1 Da to ±0.6 Da(2) From ±0.2 Da to ±1 Da	Wrong quadrupole calibration	N_samp_ × 4
Intensity alteration (E)	All signal intensities are either reduced or enhanced by a specified value	±5% (5%)	Problems with ESI source or detector gain variations	N_samp_ × 2

* One timepoint = 0.03 min.

**Table 3 metabolites-12-00993-t003:** Results of the Apiaceae plant part classification by SVM after data augmentation.

Augmentation Type	Precision (mean ± SD) *	Recall (mean ± SD)	F1-Score (mean ± SD)
No augmentation	0.72 ± 0.07	0.68 ± 0.08	0.68 ± 0.08
Chromatogram stretching (A)	0.75 ± 0.10	0.72 ± 0.12	0.73 ± 0.12
Gradient stretching (B)	0.74 ± 0.07	0.70 ± 0.10	0.70 ± 0.09
Gradient shrinkage (C)	0.73 ± 0.06	0.70 ± 0.09	0.69 ± 0.10
Mass shifts (D)	0.74 ± 0.04	0.69 ± 0.06	0.70 ± 0.05
Intensity alteration (E)	0.72 ± 0.01	0.68 ± 0.03	0.68 ± 0.03
Full augmentation (A–E)	0.77 ± 0.02	0.75 ± 0.03	0.74 ± 0.03

* Calculated from five cross-validation runs.

**Table 5 metabolites-12-00993-t005:** Results of the classification by different methods.

Results for the Whole Dataset			
Metric	SVM	CNN	SNN
Precision	0.77 ± 0.02	0.81 ± 0.05	0.81 ± 0.03
Recall	0.75 ± 0.03	0.77 ± 0.06	0.78 ± 0.04
F1	0.74 ± 0.03	0.76 ± 0.07	0.77 ± 0.05
**F1-score for plant parts classes**			
**Part**	**SVM**	**CNN**	**SNN**
Roots (33) *	0.55	0.75	0.76
Stems (45)	0.89	0.83	0.89
Leaves (60)	0.94	0.94	0.88
Fruits (48)	0.71	0.77	0.71

* Number of samples with the chosen label in the dataset before augmentation.

## Data Availability

Code and mzml files are available for download from link mentioned in Section 2.5.

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
