# Peer review of "Analysis of Primary Liquid Chromatography Mass Spectrometry Data by Neural Networks for Plant Samples Classification"

_metabolites, 2022, doi:10.3390/metabo12100993_

Round 1

Reviewer 1 Report

All my suggestions are given as a comments in the pdf document attached.

Author Response

Response to First Reviewers’ Comments

Comments 1-3. It would be better if it were high resolution, but let's see how it goes with low. After all, it is about HRMS analysis, so why does it say only low resolution in the abstract? And why was the same not done with HRMS data from Bruker? Why are 2 LC/MS devices used in this experiment at all?

Reply: We thank the reviewer for the interest to our manuscript, helpful comments and suggestions. We are working with both methods, but in this study, we tried to evaluate the opportunities of deep learning methods to extract relevant information from almost “raw” LC-MS data. This required quite much computational time even for low resolution data, but we will continue this work in future. In this work, LC-HRMS was applied only for structural elucidation of characteristic compounds, which were selected on the basis of LC-LRMS data analysis.

Is this data from the HRMS device? Was the MS/MS recorded in HRMS mode, because somewhere the correct mass was entered, and somewhere not?!

Reply Thanks for pointing this out. We have corrected several masses in the Table 4 and also changed [M+Na]+ to [2M+Na]+ etc. for compounds 3 and 4. For the neutral losses exact masses were omitted for simplicity of data representation. All high resolution MS/MS spectra are presented in the supplementary data (Fig. S2-S18).

Only 8 compounds that have not been fully identified? Seems very small? Display some chromatogram. I suggest to characterize these extracts in more detail. In the Supplementary data there are some duplicated chromatograms, but nothing special is visible there.

Reply We agree with this comment. Indeed, it was very challenging to reveal true group-characteristic markers from the highly complex and variable mixture of signals. We have added a new section (3.4), where we compare the chemical composition of studied plant extracts. It was demonstrated that coumarin markers found in leaves of these plants in our previous study were not specific enough to form a classification basis for other plant parts. In contrast, the eight markers found by means of SVM were significantly more characteristic. In addition, we have added some TICs in the supplementary file (Fig. S19-S22) to demonstrate the high variability within plant part sample groups, which had not allowed us to extract more of characteristic features.

Reviewer 2 Report

I would like to appreciate all the authors for their great contribution to understanding metabolomics data. The classification of primary and secondary metabolites is of great importance as it helps elucidate the biochemical pathways in plants.

This article will also be an important contribution to the journal in terms of understanding plant metabolism and metabolic networking.

Author Response

Response to Second Reviewers’ Comments

I would like to appreciate all the authors for their great contribution to understanding metabolomics data. The classification of primary and secondary metabolites is of great importance as it helps elucidate the biochemical pathways in plants.

This article will also be an important contribution to the journal in terms of understanding plant metabolism and metabolic networking.

Reply: We deeply thank the reviewer for this kind evaluation of our manuscript.

Reviewer 3 Report

The paper is interesting since it demonstrates to reveal specific plant part features from one family, and preserve maximum information content using SVM and NN. Why authors used only 4 data augmentation variants, still more can not be used for this study? 

Nevertheless, since no relative abundance of the metabolites exhibiting possible biological effect results from  LC-MS analysis, the authors should try to give at least some semi-quantitative data related to their content. A total ion chromatogram indicating the relevant peaks for the identified metabolites could help to give an estimate if a more specific quantitative analysis is unavailable.

Abstract: remove italics for the Apiaceae. In several other places as well. Only scientific names should be italicized, not family name. In the abstract, SVM, NN all need to be expanded at the first instance.

In the abstract, 20 plant, but in conclusion, 23 are given. In Methods, not mentioned about how many numbers, but different numbers are provided for roots, fruits, stem, etc. Keep all same consistently, it will be confusing for the readers.

Table 4 citations show from 39 to 47. I couldn't find 37,38 before that table, check it.

Author Response

Response to Third Reviewers’ Comments

The paper is interesting since it demonstrates to reveal specific plant part features from one family, and preserve maximum information content using SVM and NN. Why authors used only 4 data augmentation variants, still more can not be used for this study? 

Reply: We thank the reviewer for all the constructive comments and advices. We appreciate an opportunity given to us to improve our manuscript. Typically, one or two data modification variants are employed for data augmentation [20,27]. In this work, we have tried to mimic real changes in the experimental conditions in order to simultaneously enhance the data and evaluate the robustness of the approaches. As a result, fully augmented data allowed to achieve slightly increased F1-score values, while the accuracies were close for non-augmented data and all four partially augmented data, which demonstrated the sustainability of the classification algorithms to the modeled variations in experimental conditions. During revision we have added 1 more variant of augmentation. Its description is added in the Table 2.

Nevertheless, since no relative abundance of the metabolites exhibiting possible biological effect results from LC-MS analysis, the authors should try to give at least some semi-quantitative data related to their content. A total ion chromatogram indicating the relevant peaks for the identified metabolites could help to give an estimate if a more specific quantitative analysis is unavailable.

Reply: We agree with this comment. In fact, the relative ion abundancies were used as an input in our models because data was normalized during preprocessing. However, due to complexity of the studied materials from 20 different planet species, most of the features had drastic variations in signal intensity, therefore quantitative analysis is hindered. (In fact, the challenge was in overcoming the species-related variations and discover real plant part related markers). Moreover, the selected class-important features characteristic for plant part sample groups were corresponding to the relatively minor compounds, which were hidden in the noise on the TICs (Fig. S19-S22). Even on XIC chromatograms their peaks could be observed between other visible signals (Fig. S1). However, we added some discussion about their relative abundances in plant samples in the added Section 3.4 and added Fig. 3 for illustration.

Abstract: remove italics for the Apiaceae. In several other places as well. Only scientific names should be italicized, not family name. In the abstract, SVM, NN all need to be expanded at the first instance.

Reply: Thanks for this correction. The corresponding changes were made throughout the text.

In the abstract, 20 plant, but in conclusion, 23 are given. In Methods, not mentioned about how many numbers, but different numbers are provided for roots, fruits, stem, etc. Keep all same consistently, it will be confusing for the readers.

Reply: Thanks for this suggestion. There were 23 plants from 20 different species. We have clarified this in the text:

“Apart from additional parts of 19 earlier studied plants [26], it included fruits and roots of Prangos chelantofolia, two new plant samples of Bilacunaria microcapra and one additional plant sample of Ferulago phialocarpa. Therefore, there was a set of 23 plants from 20 different plant species.”

We also added the short comment about the numbers for root, fruits, stems, and leaves samples in addition to the Table 1.

“Thus, final set of samples contained 11 roots, 15 stems, 20 leaves and 16 fruits & flowers (Table 1).”

Table 4 citations show from 39 to 47. I couldn't find 37,38 before that table, check it.

Reply: Thanks for pointing this out. We have moved the first mention of Table 4 in the proper place of the text after these references.

Round 2

Reviewer 3 Report

The developed approach was applied to distinguish different plant parts from same Apiaceae species, revise as family.

Table 1. List of subjects used in this study: 20 different species and 23 plant parts (Apiaceae) 

Is it right? Can you revise like this? Not 23 plants, it is 23 plant parts, revise accordingly. 

Conclusion

Eight differentiation markers for distinguishing the roots, stems, leaves, and fruits of 20 species of Apiaceae....

Author Response

The developed approach was applied to distinguish different plant parts from Apiaceae species, revise as family.

Reply: Corrected.

Table 1. List of subjects used in this study: 20 different species and 23 plant parts (Apiaceae). Is it right? Can you revise like this? Not 23 plants, it is 23 plant parts, revise accordingly.

Reply: Thanks for pointing this out. There were 11 roots, 15 stems, 20 leaves and 16 fruits & flowers (62 plant parts samples in total). We have corrected this in the Table 1 and throughout the text.

Conclusion: Eight differentiation markers for distinguishing the roots, stems, leaves and fruits of 20 species of Apiaceae….

Reply: Corrected.
